# Evaluation of the Effectiveness of Sex Determination in the Great Cormorant (*Phalacrocorax carbo*) Based on External Measurements

**DOI:** 10.3390/ani14162389

**Published:** 2024-08-17

**Authors:** Włodzimierz Meissner, Michał Goc, Grzegorz Zaniewicz

**Affiliations:** Department of Vertebrate Ecology and Zoology, Faculty of Biology, University of Gdańsk, Wita Stwosza 59, 80-308 Gdańsk, Poland; michal.goc@ug.edu.pl (M.G.); grzegorz.zaniewicz@ug.edu.pl (G.Z.)

**Keywords:** biometrics, sexing, *Phalacrocoracidae*, discriminant analysis

## Abstract

**Simple Summary:**

The great cormorant has become a model taxon in the study of the ecology and ecotoxicology of fish-eating species. In this study, we used 157 adult and 222 juvenile dead individuals sexed by dissection to provide a method of sexing this species using external measurements. Males were larger than females in all linear body measurements, but there were no significant differences between adults and juveniles. Thus, in further analyses data on juveniles and adults within a sex was combined. The discriminant equations provided, with wing length and bill length, provided high accuracy, with 6.2% of individuals misclassified. The method presented here makes it possible to account for sex-specific patterns in ecological studies of the great cormorant and may be applied to data collected in the past.

**Abstract:**

The possibility of sex identification of birds has substantial importance for studies on different aspects of bird ecology and behaviour. Using discriminant functions is becoming increasingly popular in studies of bird species that are monomorphic in plumage characteristics because they are cheap, hardly invasive and may be applied to data collected in the past. In this paper, we provide a discriminant function to sex great cormorants using external measurements. Males were larger than females in all linear body measurements, but there were no significant differences between adults and juveniles. Thus, data on juveniles and adults within a sex was combined. Discriminant equations with the most commonly used linear measurements, wing length and bill length, were provided. If identifying birds with discriminant function values D_2_ < −1.256 as females and those with D_2_ > 0.916 as males, 99% of birds will be correctly sexed. The method presented here makes it possible to account for sex-specific patterns in ecological studies of the great cormorant and may be applied to data collected in the past. The cross-application of discriminant functions developed for other populations of the great cormorant produces a 5.4% and 7.5% misclassification rate for birds from northern Poland using discriminant equations developed for populations in Greece and the Netherlands, respectively.

## 1. Introduction

The ability to determine the sex of individuals is important in studies on various aspects of avian biology and ecology e.g., [1,2,3,4]. Differences between sexes are well described for birds that show clear sexual dimorphism in plumage characteristics, but there are a lot of species where the sexes do not differ in appearance, which obstructs studies of sex-specific patterns [5]. In cormorants *Phalacrocoracidae*, sexes may differ in various aspects of their biology and ecology, such as migration distance [6,7], diving behaviour [8,9], diet [10,11,12] and the criteria determining breeding site fidelity [13]. This family consists of about 40 medium-to-large species of diving fish-eaters [14]. They are monomorphic in plumage characteristics, but males are larger than females [14], which allows for the use of a discriminant function derived from the biometry of individuals with known sex for sex determination. This method is cheap, hardly invasive and may be applied to data collected in the past, so it is becoming increasingly popular in studies of monomorphic bird species e.g., [15,16,17,18,19]. Alternatively, DNA-based sexing methods [20,21] may be used, which allow researchers to determine a bird’s sex accurately, but this procedure is costly, especially for large samples.

Some papers provide discriminant functions for sexing different cormorant species [22,23,24,25,26], including the great cormorant *Phalacrocorax carbo* [27,28]. However, these discriminatory equations were developed for local populations. Their effectiveness has not been verified on other populations of these species as the accuracy of sexing using discriminant functions may be low in species with a wide geographical range or species exposed to different local selection pressures [29,30,31]. Furthermore, some linear measurements may change with an individual’s age, forcing the use of different discriminating equations for various age classes [32,33], which also should be considered when comparing linear measurements of adult and juvenile birds.

In the western Palearctic, the great cormorant inhabits a vast area stretching from the British Isles and France through Switzerland to northern Italy and across all other European countries, including Scandinavia and northwestern Russia to the east, extending further to the Black and Caspian Seas [14]. There are two subspecies in this area, the ‘Atlantic’ *P. c. carbo* and the ‘continental’ *P. c. sinensis*. The former is mainly found on the seacoasts of western and northern Europe, while the latter inhabits the area around the Baltic Sea and the mid-continent [14,34]. It is classed as a conflict species due to reported losses of fish stocks on fishponds and in natural ecosystems [35,36]. Because of its widespread occurrence, the great cormorant has become a model taxon in the study of the ecology and ecotoxicology of fish-eating species [37,38,39,40].

In this paper, we aimed to provide a method of sexing the great cormorant using external measurements. We also validate the sexing method provided for the same species in Greece [28] and in the Netherlands [27] using our data from birds collected in northern Poland. In this way, we want to determine whether they show convergent results and if the method can be applied to birds from different parts of their European range.

## 2. Materials and Methods

Birds were collected from July to December between 2006 and 2022 in the Kashubian Lake District in northern Poland. The sample consisted of 157 adult and 222 juvenile (first year of life) great cormorants shot in fishponds under a permit granted by the Directorate of Environmental Protection in Gdańsk.

### 2.1. Ageing and Morphometric Measurements

The age of great cormorants was determined based on plumage characteristics, distinguishing juveniles (in their first year of life) and adults (older than one year) [34]. Sex was determined by examining gonads through dissection. Wing length (maximum flattened chord) was measured with a ruler (1 mm accuracy), while the total head, tarsus and bill lengths and both bill depths were measured using callipers (0.1 mm accuracy) [41]. Bill length was measured from tip to feathering; maximum bill depth was measured at gonys, while minimum was measured at the narrowest part of the bill (Figure 1). Not all biometric data were taken from every individual, so the sample size varied for some measurements. The body mass was omitted in analyses because it varies considerably during a year due to the accumulation of fat reserves [27], as well as oesophageal and gastric contents or feather moisture. Outer primary feathers were carefully checked for any basal sheaths to ensure feather growth was complete and there was no conspicuous wear of the wing tip; thus, only individuals with the primary moult completed and an unworn tip of the longest primary were measured. All birds were measured by only one person (Michał Goc); hence, it is assumed that there are no potential differences in measurements taken by different people.

### 2.2. Statistical Analysis

Differences in morphological traits between males and females were tested with a two-sample *t*-test [42]. The size dimorphism index (SDI) was determined by Lovich and Gibbons’ sexual dimorphism index (SDI) [43], in which mean values of linear body measurements of both sexes are taken into account. However, we modified the original equation according to Dorofeev et al. [33] and the result is presented as a percentage.
(1)$$SDI=\left[\left(\frac{mean\;value\;of\;males}{mean\;values\;of\;females}\right)-1\;\right]*100$$

Therefore, positive values indicate a higher mean for a given measurement in males (larger sex) and negative values indicate this in females.

A backward stepwise general discriminant analysis (GDA) was conducted to determine which variables best classified the sex of great cormorants. The inclusion of a measurement in the model was based on Wilk’s lambda ratio, with a minimum partial F to enter the model equal to 3.84 and a maximum partial F to be removed from the model equal to 2.71. As in other similar studies, a priori classification probabilities are equal for both sexes (*p* = 0.50) e.g., [32,44,45]. Despite the presence of a male-biassed sex ratio in the sample, we did not adjust the prior probabilities for the analyses because we did not have prior knowledge of the sex ratio in the studied population.

As in other papers [32,33,46], in the discriminant function analysis (DFA), the sexes were coded ‘−1’ for females and ‘1’ for males and the equations presented in this paper are based on unstandardized canonical discriminant function coefficients, with a discriminant score D < 0 indicating a female and D > 0 indicating a male. Standardised canonical coefficients were also given to assess the contribution of one predictor in the context of the other predictors in the model. We validated the success rate of classification of each discriminant function with squared Mahalanobis distances from each sex-group centroid, where the given individual is classified into the sex group for which it has the highest posterior classification probability [47]. To show the degree of overlap between sexes in the two most dimorphic measurements, 95% prediction intervals were presented; these are a range of values likely to contain the value of any single new observation given the settings of the predictors [48]. The statistical analyses were performed using Statistica 13.3 software (TIBCO Software Inc., Palo Alto, CA, USA).

## 3. Results

There were no significant differences found between adult and juvenile females in the mean values of all measurements (*t*-test, *p* > 0.100 in all cases). Adult males have significantly larger maximum bill depth than juveniles (*t*-test, t = 2.56, *p* = 0.011), but the difference between means is only 0.3 mm. Other linear measurements do not differ between adult and juvenile males (*t*-test, *p* > 0.063 in all cases). Thus, in further analyses data on juveniles and adults within a sex were combined, as in other papers analysing biometrics of the great cormorant [27,28]. All data met the assumptions of the homogeneity of variance (Brown–Forsythe test, *p* > 0.292) and normality (Shapiro–Wilk test, *p* > 0.078), except for wing length in males (Brown–Forsythe test, F = 5.65, *p* = 0.018) and bill length and minimum bill depth in females (Shapiro–Wilk test, W = 0.96, *p* = 0.018 and W = 0.91, *p* < 0.001, respectively). Nevertheless, we did not transform this variable because discriminant analysis (DFA) robustly withstands deviations from normality, especially for large sample sizes [42,49].

Males were larger than females in all linear body measurements (Table 1). The most sexually dimorphic trait is minimum bill depth with SDI 13%, but the overlap between the sexes is large (Figure 2). Hence, only extremely large or small individuals may be safely sexed using single measurements. When taking into account minimum bill depth and bill length simultaneously, 29% of males and 22% of females were found within the overlapping zone of two ellipses showing a 95% confidence interval for these two most dimorphic traits (Figure 3). The wing length revealed the lowest coefficient of variation both in males and females, much lower than all bill measurements (Table 1).

In the stepwise procedure, wing length, bill length and minimum bill depth were selected (Table 2, equation D_1_). We also present a model with wing and bill length (Table 2, equation D_2_), as these measurements are most often taken in bird studies [41,50]. Wing length contributes the most to both models with standardised canonical coefficients of 0.584 and 0.703 for equations D_1_ and D_2_, respectively. Two equations have high and very similar accuracies of almost 94% for correctly sexed birds (Table 2). Thus, the inclusion of the minimum bill depth in the model did not significantly increase the proportions of birds sexed correctly (Table 2) and we recommend the D_2_ equation for further studies, especially as the measurement of minimum bill depth in cormorants has not been standardised see [22,23,51].

The overlapping range between sexes in D_2_ score occurred mostly between −1.5 and 1.5 (Figure 4). If identifying birds with D_2_ < −1.256 as females and those with D_2_ > 0.916 as males, only 1% of males and 1% of females will be misclassified (Figure 4). However, when applying this criterion to the 275 great cormorants used in this study, a large proportion—38% of the individuals—fell within this range and remained unsexed.

## 4. Discussion

Although male and female cormorants show no differences in plumage characteristics, sex determination in most of them is possible using a combination of linear measurements. The accuracy of the discriminant functions used to determine the sex of different cormorant species is high. The misclassification rate is always less than 15% [41,50] or even 5% [22,24,28,51,52]. Body mass has been identified in some studies as showing a statistically significant difference between the sexes [27,28,51] and has even been included in the discrimination equation [22]. Still, it is strongly dependent on accumulated fat stores, which tend to fluctuate throughout the year. For this reason, body mass is excluded from discrimination equations in cormorants [24,28], but see [22] and, for other species, [53,54,55].

The most sexually dimorphic traits in the majority of cormorant species are various bill measurements [22,24,26,51,56], which was also confirmed in the great cormorant [27,28], this study. This leads to sex differences in diet, as cormorant males eat higher proportions of large fish species than females [11,12,57,58]. Wing length is another measurement that has often been included in discriminant functions as the best one for determining the sex of cormorants [23,24,25,52]. This trait presented the lowest coefficient of variation among all linear measurements in different cormorant species [28,51], this study, often becoming the most accurate single predictor of sex. In cormorants, males are heavier than females [22,24,28,51] and longer wings result in a larger wing area, which is necessary to maintain high flight efficiency with greater body mass. In cormorants, males forage deeper than females [8,9,10]. Hence, the size of the legs, which cormorants use as a driving force when diving, also differentiates well between the sexes, as has been confirmed for several species when tarsus length was included in the discriminant equation [24,28,51]. In our sample, males had a longer tarsus than females, but this measurement did not have sufficiently high discriminatory power to enter the model, as was the case for birds from Greece [28].

The sexing method provided for individuals from Greece [28] and the Netherlands [27] performed well for birds from northern Poland. Correct sex assignment rates varied only slightly between these regions (Table 3), which reflects small differences in the size of sexual dimorphism in a large part of this subspecies’ range. European subspecies of the great cormorant can be separated relatively clearly according to the areas they inhabit [59]. Hence, the vast majority of individuals from Greece, the Netherlands and Poland belong to the *P. c. sinensis* subspecies, as only this subspecies inhabits these countries [34,60]. Although there is an ecological segregation of the two subspecies [59,61,62] and *P. c. carbo* birds are mainly sedentary, while *P. c. sinensis* are highly migratory [63,64], the presence of *P. c. carbo* at least in samples from the Netherlands and Poland cannot be ruled out, as indicated by ringing recoveries [65,66]. The two subspecies differ in linear dimensions, where *P. c. carbo* has a longer bill and greater bill depth [60]. The high convergence of the cross-validation results of the three discriminant equations for this species indicates that in all these analyses the contribution of subspecies *P. c. carbo* was rather low, or the proportion of both subspecies is similar. However, the location of the data collection sites for our publication indicates that the possible contribution of individuals from this subspecies was small. Thus, discriminant equations from this study may be applied to great cormorants from the *P. c. sinensis* subspecies.

Both discriminant equations provided in this study performed slightly better for males than females. This is related to the difference in size distribution by sex, as proportionally more large females were assigned as males and fewer small males were assigned as females. Hence, using the equations provided in this study may produce sex bias; however, to substantially limit the error, individuals with discriminant scores D_2_ between −1.256 and 0.916 should be treated as unsexed.

## 5. Conclusions

In the great cormorant, males are larger than females in all linear body measurements, but there are no significant differences between adults and juveniles. For the sex determination of this species, we recommend two measurements, wing length and bill length, because the way to measure another highly dimorphic trait, bill depth, is not standardised. The discriminant function presented here allows one to sex birds from the subspecies *P. c. carbo* with high accuracy and makes it possible to account for sex-specific patterns in ecological studies of the great cormorant.

## Figures and Tables

**Figure 1 animals-14-02389-f001:**
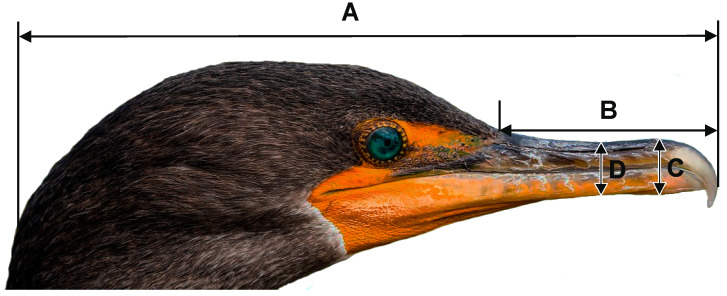
Head and bill measurements taken on the great cormorant. A—the total head length, B—bill length, C—maximum bill depth, D—minimum bill depth.

**Figure 2 animals-14-02389-f002:**
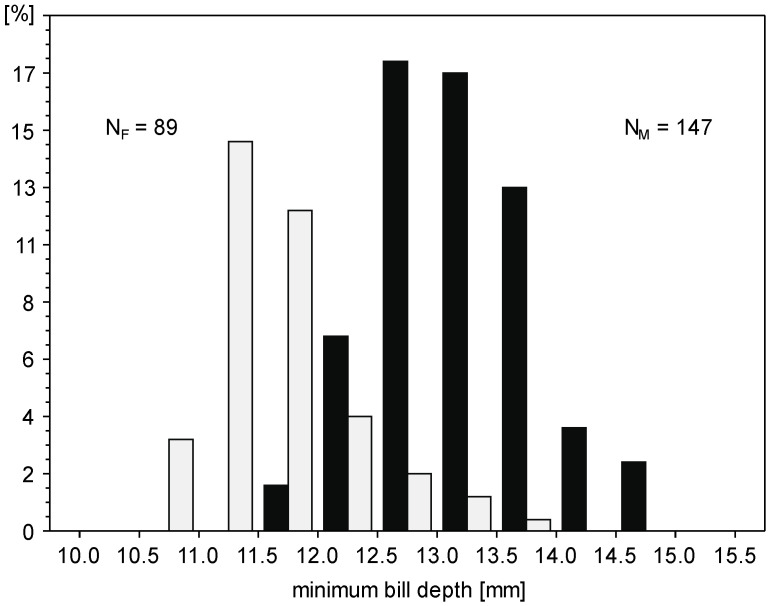
Distribution of minimum bill depth of great cormorant males (black) and females (light grey). The sample sizes of males (N_M_) and females (N_F_) were given.

**Figure 3 animals-14-02389-f003:**
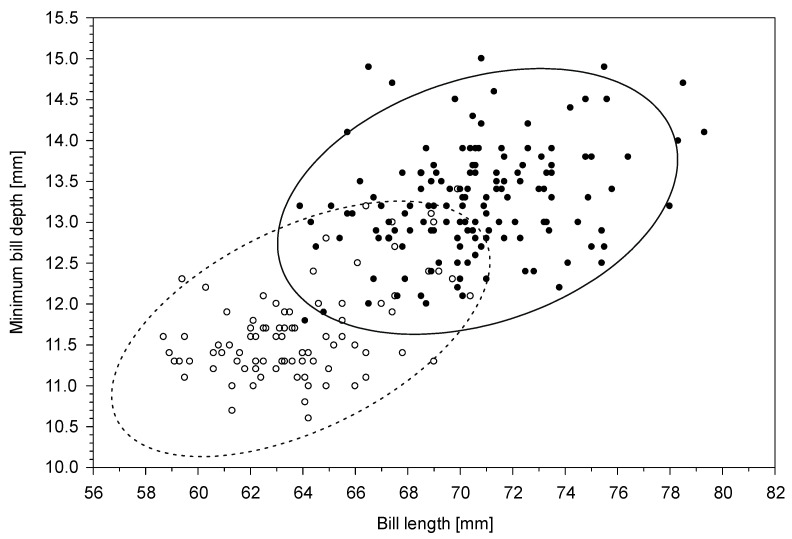
The relationship of the bill length and minimum bill depth in males (black dots) and females (white dots) of the great cormorant against the bill length. The ellipses show the 95% prediction intervals for a single observation, given the parameter estimates for the bivariate distribution computed from the data for males (solid line) and females (dashed line).

**Figure 4 animals-14-02389-f004:**
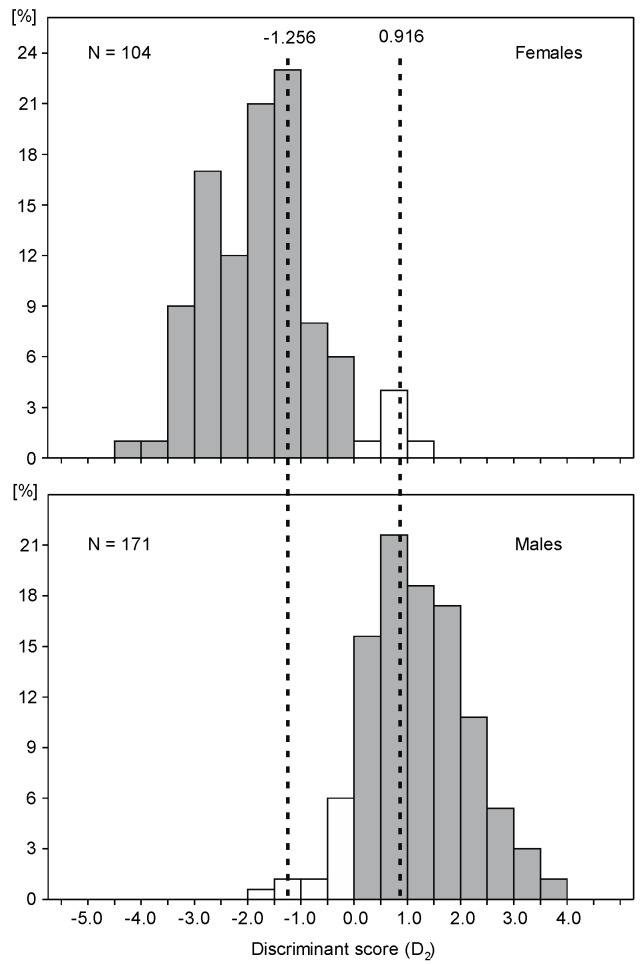
Distribution of the discriminant score D_2_ calculated for males and females sexed by dissection. Grey and white bars are for correct and incorrect classifications. Dashed lines show D_2_ border values of −0.1256 and 0.916, which allowed for 99% correct classifications of females and males.

**Table 1 animals-14-02389-t001:** Differences in mean linear measurements between males and females of the great cormorant, with the coefficient of variation (V) and the modified sexual dimorphism index (SDI) according to Lovich and Gibbons [43].

Measurement [mm]	Males			Females			*t*-Test		SDI
	Mean	V	N	Mean	V	N	t	*p*	
Wing length	352.8	2.1%	218	332.2	2.7%	139	24.24	*p* < 0.001	6.2%
Bill length	70.60	4.4%	188	63.53	5.2%	109	18.97	*p* < 0.001	11.1%
Maximum bill depth	14.51	5.5%	188	12.91	6.9%	111	15.78	*p* < 0.001	12.4%
Minimum bill depth	13.26	4.9%	147	11.70	5.2%	89	18.16	*p* < 0.001	13.3%
Tarsus length	69.19	3.8%	109	64.73	3.8%	72	11.51	*p* < 0.001	6.9%

**Table 2 animals-14-02389-t002:** Equations for calculating discriminant scores. The percentage of birds sexed correctly is given according to squared Mahalanobis distances from each sex-group centroid. WL—wing length, BL—bill length, MBD—minimum bill depth.

Equation	Correctly Sexed		
	Males	Females	All
D_1_ = 0.073·WL + 0.082·BL + 0.785·MBD − 40.623	95.4%	91.5%	93.9%
D_2_ = 0.087·WL + 0.161·BL − 40.843	96.3%	91.3%	93.8%

**Table 3 animals-14-02389-t003:** Proportions of great cormorants from northern Poland sexed correctly using discriminant functions developed for populations from Greece [28] and the Netherlands [27]. WL—wing length, BL—bill length, MBD—minimum bill depth, TL—tarsus length.

Population	Equation	Correctly Sexed		
		Males	Females	All
Greece	D = 0.073·WL + 0.123·BL + 0.310·TL − 53.693	94.2%	95.1%	94.6%
The Netherlands	D = 0.086·WL + 0.148·BL − 39.869	89.6%	96.6%	92.5%

## Data Availability

The data presented in this study are available at https://doi.org/10.18150/DFKJHB.

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
