# Peer review of "Evaluation of the Effectiveness of Sex Determination in the Great Cormorant (Phalacrocorax carbo) Based on External Measurements"

_animals, 2024, doi:10.3390/ani14162389_

Round 1

Reviewer 1 Report

Comments and Suggestions for Authors

This manuscript presents an interesting study on sexual dimorphism in Great Cormorants from Poland. The manuscript is generally well-written, and the analytical approach is well-conceived and executed. However, I have some suggestions and a few concerns that I hope will be useful for the authors in refining their work.

The authors should clarify what their study contributes beyond the existing literature. They mention that previous data cannot be generalized and are only applicable to specific populations. Please provide a rationale for how and why the data from this study advances the knowledge of sexual dimorphism in Great Cormorants.

The section on currently recognized subspecies of Great Cormorants is mentioned briefly and is somewhat confusing. Since the study focuses on a well-known species, it would be beneficial to explicitly state the hypotheses and predictions. In addition, please elaborate on the reasons for comparing your data with those from Greece and the Netherlands.

Ethical Considerations: It is essential to address ethical considerations in this study. Since all birds used in this study were shot, the authors should provide a clear ethical statement, including details on permits and authorizations from relevant ethics committees. This information should be clearly stated in the Methods section, along with the respective process and authorization numbers.

Specific Comments:

Line 39: Family-group taxa names should not be italicized.

Line 42: The sentence is awkward. Clarify that sexes in cormorants are alike in terms of plumage, but males are somewhat larger (and heavier?) than females. Discriminant analysis has been used to separate sexes in some species.

Line 51: The current equation was proposed for specific populations, and due to possible geographical variation in measurements, it cannot be extrapolated to other populations across the species' distribution. Please elaborate.

Lines 61-67: Clarify whether your data refer only to the nominate subspecies. It was unclear which subspecies the birds from Greece belong to and why this comparison was made. If you prefer not to elaborate on this topic, please clarify that your data/comparisons refer only to coastal populations.

Line 68: If existing equations/methods for sexing Great Cormorants are already available, clarify that you are expanding the range of these methods rather than proposing a new method.

Lines 74-77: For ethical reasons, provide a clear statement regarding authorization for shooting birds and other pertinent permits here.

Line 79-80: If the birds were divided into two age classes, briefly describe these classes rather than simply providing a reference.

Line 81: Clarify that sex identification was done by examining gonads through dissection.

Line 188: While cormorants are monomorphic in plumage, they do exhibit sexual dimorphism in size and weight. Clarify whether this qualifies as sexual dimorphism.

Line 217: There is no need to state the subspecies for Phalacrocorax carbo sinensis, as the trinomen already indicates this taxonomic category. Also, do not abbreviate the specific epithet, as ICZN does not recommend this.

Line 218, 220, 224: Phalacrocorax carbo carbo should be in italics, and the specific epithet should not be abbreviated.

Comments on the Quality of English Language

Overall, English is good. Nevertheless, I recommend further review after the new, revised version is submitted.

Author Response

The authors should clarify what their study contributes beyond the existing literature. They mention that previous data cannot be generalized and are only applicable to specific populations. Please provide a rationale for how and why the data from this study advances the knowledge of sexual dimorphism in Great Cormorants.

This is explained in the last paragraph of the Introduction, where we add a sentence to clarify this.

The section on currently recognized subspecies of Great Cormorants is mentioned briefly and is somewhat confusing. Since the study focuses on a well-known species, it would be beneficial to explicitly state the hypotheses and predictions. In addition, please elaborate on the reasons for comparing your data with those from Greece and the Netherlands.

We have added a sentence at the bottom of the last paragraph of the Introduction that clarifies this.

Ethical Considerations: It is essential to address ethical considerations in this study. Since all birds used in this study were shot, the authors should provide a clear ethical statement, including details on permits and authorizations from relevant ethics committees. This information should be clearly stated in the Methods section, along with the respective process and authorization numbers.

Details of these permits have been added.

Line 39: Family-group taxa names should not be italicized.

It was corrected.

Line 42: The sentence is awkward. Clarify that sexes in cormorants are alike in terms of plumage, but males are somewhat larger (and heavier?) than females. Discriminant analysis has been used to separate sexes in some species.

It was corrected: “They are monomorphic in plumage characteristics, but males are larger than females…..”

Line 51: The current equation was proposed for specific populations, and due to possible geographical variation in measurements, it cannot be extrapolated to other populations across the species' distribution. Please elaborate.

This is explained there: “However, these discriminatory equations were developed for local populations. Their effectiveness has not been verified on other populations of these species as the accuracy of sexing using discriminant function may be low in species with a wide geographical range or exposed to different local selection pressures.”

Lines 61-67: Clarify whether your data refer only to the nominate subspecies. It was unclear which subspecies the birds from Greece belong to and why this comparison was made. If you prefer not to elaborate on this topic, please clarify that your data/comparisons refer only to coastal populations.

The subspecies affiliation of birds used for the analyses in this manuscript and in the publications from Greece and the Netherlands is discussed in the Discussion section (from line 216). Therefore, we did not want to write about it in this part of the manuscript, as the assessment of the species affiliation comes from the biometric analysis, the results of which appear in the Results chapter. Certainly, the majority of individuals belonged to subspecies sinensis, but the presence of a few individuals from subspecies carbo cannot be excluded.

Line 68: If existing equations/methods for sexing Great Cormorants are already available, clarify that you are expanding the range of these methods rather than proposing a new method.

This is explained in the last paragraph of the Introduction: “…we aimed to provide a method of sexing the great cormorant using external measurements. We also validate the sexing method provided for the same species in Greece [28] and in the Netherlands…”

Lines 74-77: For ethical reasons, provide a clear statement regarding authorization for shooting birds and other pertinent permits here.

Details of these permits have been added.

Line 79-80: If the birds were divided into two age classes, briefly describe these classes rather than simply providing a reference.

The age of the birds was determined by plumage features, as mentioned in the text. In my opinion, describing these details goes beyond the subject of this manuscript. Especially as no statistically significant differences in body dimensions were found between these age classes.

Line 81: Clarify that sex identification was done by examining gonads through dissection.

It was corrected: "Sex was determined by examining gonads through dissection."

Line 188: While cormorants are monomorphic in plumage, they do exhibit sexual dimorphism in size and weight. Clarify whether this qualifies as sexual dimorphism.

It was corrected: “Although male and female cormorants show no differences in plumage characteristics, sex determination in most of them is possible using a combination of linear measurements.”

Line 217: There is no need to state the subspecies for Phalacrocorax carbo sinensis, as the trinomen already indicates this taxonomic category. Also, do not abbreviate the specific epithet, as ICZN does not recommend this.

Instead of full trinomen (e.g. Phalacrocorax carbo sinensis) I used abbreviations like P. c. sinensis because such abbreviations are widely used in ornithological journals.

Line 218, 220, 224: Phalacrocorax carbo carbo should be in italics, and the specific epithet should not be abbreviated.

It was corrected, but I left the abbreviation of the full trinomen as I explained above.

Reviewer 2 Report

Comments and Suggestions for Authors

I do not think this manuscript could be published in Animals.

See attached comments.

Author Response

While I am thinking data in this study is no problem, I found methods they suggested may not be applied to active data collection, especially in the field.

Cormorants can be caught in the field using a variety of methods, and the ability to sex them adds more options for data analysis. Our manuscript is not limited to providing discriminant functions for sexing birds, but also presents and discusses data on the magnitude of sexual dimorphism in this species and validates other equations published in other papers.

It is also not “cheap” or “invasive”, because any researcher should capture a lot of birds to measure body index. But if the researcher can capture birds, then sex determination of these birds should be easy to do and should do using method of 100% accuracy (e.g. Griffiths et al. 1998).

Indeed, molecular sexting provides 100% accuracy, but is expensive and time-consuming. Therefore, discriminant functions are widely used to sex birds in a non-invasive way (without the need for blood or feather samples) and without the costs associated with laboratory procedures.

In addition, the method of this paper was still with 6.2% of individuals misclassified.

The misclassification rate of 6.2% is quite low when comparing this figure with the results of similar studies on cormorants or other waterbirds. When working with a reasonable sample size of captured birds, it is not a problem to miss some of them in further analyses. There are published articles that easily deal with this. 

In addition, the method of this paper was still with 6.2% of individuals misclassified.

The misclassification rate of 6.2% is quite low. However, we also present criteria in the manuscript to achieve a 1% misclassification rate.

Thus, I do not think this study could be published in Animals. I must reject it any way, as this study is not helpful, not applied in action.

I cannot agree with this opinion. Published papers that provide discriminant functions for determining the sex of birds achieve a high citation index because scientists commonly use them in their research. Moreover, our paper compares the results of two other studies on this species and the conclusions are interesting - at least in our opinion. Of course, the final decision to publish the manuscript lies with the editors.

Pages 3-6: too many errors “Error! Reference source not found” and these made any review impossible.

We apologise for this - it is some kind of technical problem. It will be easily resolved.

Reviewer 3 Report

Comments and Suggestions for Authors

In this manuscript, authors analyze sexual-size dimorphism in Great Cormorants (Phalacrocorax carbo) from northern Poland, providing discriminant functions to sex adults and juveniles based on external measurements. Sample size is very large: 157 adults and 222 juveniles (gathered between 2006 and 2022). As expected, males were larger than females in all measurements while there were no differences between adults and juveniles. Discriminant functions provided correctly classified almost 94% of individuals. This result is in consonance with two previous studies addressing sexual size dimorphism in other Great Cormorant populations. In this sense, discriminant functions developed for Greece and the Netherlands correctly classified 92.5-94.6% of birds from northern Poland. The manuscript is clear, concise and properly analyzed. Following, I made some minor suggestions.

Line 42: “They are monomorphic , but males are larger than females …” (about cormorants). That sentence is strange because cormorants are monomorphic in plumage and appearance but dimorphic in size (the topic analyzed here). Thus, cormorants are typically considered dimorphic species. Please, correct that concept along the manuscript (i.e cormorants are dimorphic species).

Line 107: I am wondering if a forward technique provides the same result. This forward approach would prioritize simpler equations (less is more), and perhaps, no three-parameter equation would be necessary (e.g. WL-BL or WL-MBD could provide a similar classification, similar to a WL-BL-MBD equation).

Line 168: inclusion instead exclusion.

Lines 175-179: I am not seeing the advantage of an improve of 5% (from 94% to 99%) in classification rates at the expense of discarding almost 40% of the sample. I suggest the elimination of this notion (or at least, softening the relevance of that).

Lines 188-189: “Although cormorants are monomorphic, sex determination in most of them is possible using a combination of linear measurements”. If you know studies that report subtle sex-dimorphism in a cormorant species that precludes sex determination by external measurements, please cite here. Also, see comment for line 42.

Lines 192-195: I suggest the elimination of those sentences. From the studies in cormorants cited there, only the “22” reference used body mass in a discriminant equation.

Author Response

Line 42: “They are monomorphic , but males are larger than females …” (about cormorants). That sentence is strange because cormorants are monomorphic in plumage and appearance but dimorphic in size (the topic analyzed here). Thus, cormorants are typically considered dimorphic species. Please, correct that concept along the manuscript (i.e cormorants are dimorphic species).

This was corrected along the manuscript.

Line 18: Using discriminant functions is becoming increasingly popular in studies of bird species that are monomorphic in plumage characteristics

Line 42: They are monomorphic in plumage characteristics, but males are larger than females.

Line 189: Although male and female cormorants show no differences in plumage characteristics, sex determination in most of them is possible using a combination of linear measurements.

Line 107: I am wondering if a forward technique provides the same result. This forward approach would prioritize simpler equations (less is more), and perhaps, no three-parameter equation would be necessary (e.g. WL-BL or WL-MBD could provide a similar classification, similar to a WL-BL-MBD equation).

We checked this and the forward technique provided the same results with WL, BL and MBD included in the final model.

Line 168: inclusion instead exclusion.

It was corrected.

Lines 175-179: I am not seeing the advantage of an improve of 5% (from 94% to 99%) in classification rates at the expense of discarding almost 40% of the sample. I suggest the elimination of this notion (or at least, softening the relevance of that).

This sentence was changed for softening the relevance of this notion:

However, when applying this criterion to the 275 great cormorants used in this study, a large proportion—38% of the individuals—fell within this range and remained unsexed.

Lines 188-189: “Although cormorants are monomorphic, sex determination in most of them is possible using a combination of linear measurements”. If you know studies that report subtle sex-dimorphism in a cormorant species that precludes sex determination by external measurements, please cite here. Also, see comment for line 42.

I haven’t found any published source showing a lack of statistically significant differences between males and females in cormorant species.

Lines 192-195: I suggest the elimination of those sentences. From the studies in cormorants cited there, only the “22” reference used body mass in a discriminant equation.

Indeed, only in reference 22 authors used body mass in discriminant equation, but in other papers cited here [28, 51, 52] statistically significant differences in body mass of males and females were reported. In reference 24, this difference was significant, but was not tested statistically and we remove it from this sentence, which was changed to:

Body mass has been identified in some studies as showing a statistically significant difference between the sexes [27,28,51] and has even been included in the discrimination equation [22].

Reviewer 4 Report

Comments and Suggestions for Authors

Review of “Evaluation of the Effectiveness of Sex Determination in the 2 Great Cormorant (Phalacrocorax carbo) Based on External 3 Measurements:

Being able to determine the sex of a waterbird through external measurements is an important tool.  This paper has excellent sample sizes and sound analysis. There are some writing issues that I tried to address and I think the authors could do more to compare the birds measured in Poland to those in Greece and the Netherlands.  I really appreciated the photo of the cormorant with the measurements – that was very helpful.

 Here are some specific comments:

 1.     Line 35 “The ability of sex identification is important in studies on various aspects of avian 35 biology and ecology”.

Reword to something like “The ability to determine the sex of individuals”

2.     Lines 36-37. “Differences between sexes are well described for birds that show clear sexual dimorphism but there are a lot of species where the sexes do not differ in appearance, which obstructs studies of sex-specific patterns [5].”

Your research is about a species that differs significantly in size between the sexes so I’m not sure what this sentences is saying and the citation (the book) is an odd thing to cite. I think you need to clarify this section. Perhaps you are saying they are mono-chromatic? If that is true, I would find a more appropriate reference.

 3.     Lines 43-44. This is related to the point above, you write  “They are monomorphic, but males are larger than females…”.  If they are monomorphic (all the same) then they would not be a different size.  Perhaps you are saying they are monochromatic (all the same colors) but sexually dimorphic in terms of morphological characteristics?  I would clarify this. 

 4.     Lines 38-40. In cormorants (Phalacrocoracidae) sexes may differ in various aspects of their biology and ecology such as migration distance [6–7], diving behaviour [8-9], diet [10–11] and the criteria determining breeding site fidelity”. I would suggest deleting those words

 5.     Line 59, I would use the word “to” rather than “via”

 6.     Line 81, I think the word “sex” is now used rather than “gender” when writing about animals “Gender was determined by dissection”… Gender is a complicated word these days. 

 7.     Figure 3 caption “Figure 3. The relationship of the bill length and minimum bill depth in males (black dots) and 158 females (white dots) of the great cormorant against the tarsus length.”  I can’t tell if this is an error but in the text you write “When taking into account minimum bill depth and bill length simultaneously, 29% of males and 22% of females were found within the overlapping zone of two ellipses showing a 95% confidence interval for these two most dimorphic traits (Figure 3).” You don’t mention the tarsus. You either need to fix the error or clarify this in the methods and results.

 8.     Lines 166-167 “Two equations have high and very similar efficiencies of almost 94% of correctly sexed birds (Table 2).” I might use the word “accuracy” rather than efficiencies. 

9.     Lines 176-178. “If identifying birds with D2 < –1.256 as females and those with D2 > 0.916 as males, only 1% of males and 1% of females will be misclassified (Fig. 4). When using this criterion to 275 great cormorants used in this study, 38% of individuals were within this range and remained unsexed.”

 That seems like a lot of birds are unsexed. I would mention that in Figure 4 as the way it is written it seems like you have 99% accuracy overall.

 10.  Line 186. In Table 3, I would define WL and BL (which were defined before). You particularly need to define TL which is not mentioned.

11.  Line 188, “Although cormorants are monomorphic, sex determination in most of them is possible using a combination of linear measurements” .  Like in the introduction, I think monochromatic is the correct word (not monomorphic).

 12.  Paragraph that starts at line 211. 

a.      I think it is interesting to compare the cormorants in your study to those in Greece and Netherlands. I had trouble following the paragraph and wondered if a figure or table with the measurements for each area might be useful. 

b.      I also thought it was interesting that the Greece study found including TL would be useful.

c.       Several of the Latin names in this are not italicized and should be -  (particularly P. c. carbo)

 d.      Lines 237-“For the sex determination of this species, we recommend two measurements: wing length and bill length, as  because the way to measure another highly dimorphic trait, bill depth, is not standardized” small change

Comments on the Quality of English Language

It seems like the big issue is some word choice and I've tried to correct where I can.  

Author Response

  1. Line 35 “The ability of sex identification is important in studies on various aspects of avian biology and ecology”.

Reword to something like “The ability to determine the sex of individuals”

It was changed as you suggested.

  1. Lines 36-37. “Differences between sexes are well described for birds that show clear sexual dimorphism but there are a lot of species where the sexes do not differ in appearance, which obstructs studies of sex-specific patterns [5].”

Your research is about a species that differs significantly in size between the sexes so I’m not sure what this sentences is saying and the citation (the book) is an odd thing to cite. I think you need to clarify this section. Perhaps you are saying they are mono-chromatic? If that is true, I would find a more appropriate reference.

This sentence was slightly changed, and I hope it is now clear that we mean differences in plumage characteristics. The book quoted here, among other things, shows and summarises the results of various studies with differences found between the sexes. Therefore, in my opinion, it is a good literature source to cite here. Otherwise, the list of relevant references would be very long.

  1. Lines 43-44. This is related to the point above, you write  “They are monomorphic, but males are larger than females…”.  If they are monomorphic (all the same) then they would not be a different size.  Perhaps you are saying they are monochromatic (all the same colors) but sexually dimorphic in terms of morphological characteristics?  I would clarify this. 

This was changed to: They are monomorphic in plumage characteristics, but males are larger than females

  1. Lines 38-40. In cormorants (Phalacrocoracidae) sexes may differ in various aspects of their biology and ecology such as migration distance [6–7], diving behaviour [8-9], diet [10–11] and the criteria determining breeding site fidelity”. I would suggest deleting those words

I think this sentence is needed because it shows that being able to determine sex in cormorant studies is very important, since these species have sex differences in various aspects of their lives. That is why I left this sentence in the text.

  1. Line 59, I would use the word “to” rather than “via”

The whole sentence was changed.

  1. Line 81, I think the word “sex” is now used rather than “gender” when writing about animals “Gender was determined by dissection”… Gender is a complicated word these days. 

Right! It was changed as you suggested.

  1. Figure 3 caption “Figure 3. The relationship of the bill length and minimum bill depth in males (black dots) and 158 females (white dots) of the great cormorant against the tarsus length.”  I can’t tell if this is an error but in the text you write “When taking into account minimum bill depth and bill length simultaneously, 29% of males and 22% of females were found within the overlapping zone of two ellipses showing a 95% confidence interval for these two most dimorphic traits (Figure 3).” You don’t mention the tarsus. You either need to fix the error or clarify this in the methods and results.

This is simply our mistake. In the Figure 3 we use two most dimorphic traits included in discriminant function. We haven’t used the tarsus length, because it was not included in the discriminant equation. Hence we fix this error in the figure caption.

  1. Lines 166-167 “Two equations have high and very similar efficiencies of almost 94% of correctly sexed birds (Table 2).” I might use the word “accuracy” rather than efficiencies. 

It was changed as you suggested.

  1. Lines 176-178. “If identifying birds with D2 < –1.256 as females and those with D2 > 0.916 as males, only 1% of males and 1% of females will be misclassified (Fig. 4). When using this criterion to 275 great cormorants used in this study, 38% of individuals were within this range and remained unsexed.”

 That seems like a lot of birds are unsexed. I would mention that in Figure 4 as the way it is written it seems like you have 99% accuracy overall.

This sentence was changed for softening the relevance of this notion: However, when applying this criterion to the 275 great cormorants used in this study, a large proportion—38% of the individuals—fell within this range and remained unsexed.

  1. Line 186. In Table 3, I would define WL and BL (which were defined before). You particularly need to define TL which is not mentioned.

Definition of all these abbreviations was added.

  1. Line 188, “Although cormorants are monomorphic, sex determination in most of them is possible using a combination of linear measurements” .  Like in the introduction, I think monochromatic is the correct word (not monomorphic).

It was changed to: Although male and female cormorants show no differences in plumage characteristics, sex determination in most of them is possible using a combination of linear measurements.

  1. Paragraph that starts at line 211. 
  2. I think it is interesting to compare the cormorants in your study to those in Greece and Netherlands. I had trouble following the paragraph and wondered if a figure or table with the measurements for each area might be useful. 

This comparison is limited only to the sex determination of birds from Poland using the discriminant function given in papers on the great cormorant from Greece and the Netherlands. We do not have access to the data from these studies and therefore cannot carry out a more detailed analysis.

  1. I also thought it was interesting that the Greece study found including TL would be useful.

Yes, I mentioned this at the end of the previous paragraph.

  1. Several of the Latin names in this are not italicized and should be -  (particularly P. c. carbo)

It was corrected.

  1. Lines 237-“For the sex determination of this species, we recommend two measurements: wing length and bill length, as because the way to measure another highly dimorphic trait, bill depth, is not standardized” small change

It was corrected.

Comments on the Quality of English Language

It seems like the big issue is some word choice and I've tried to correct where I can.  

Thanks!

Round 2

Reviewer 1 Report

Comments and Suggestions for Authors

Overall, the authors have addressed most of my comments and suggestions very well, and I am satisfied with the revised version of their manuscript. I see no impediments to its acceptance for publication.

Comments on the Quality of English Language

The English is overall good, but a final revision is recommended. This can be safely done during the copyediting process.

Reviewer 2 Report

Comments and Suggestions for Authors

I have no further comments.